# Impact of ADA Guidelines and Medication Shortage on GLP-1 Receptor Agonists Prescribing Trends in the UK: A Time-Series Analysis with Country-Specific Insights

**DOI:** 10.3390/jcm13206256

**Published:** 2024-10-20

**Authors:** Ahmed R. N. Ibrahim, Khalid M. Orayj

**Affiliations:** Department of Clinical Pharmacy, College of Pharmacy, King Khalid University, Abha 62521, Saudi Arabia; korayg@kku.edu.sa

**Keywords:** GLP-1 receptor agonists, diabetes management, prescribing trends, medication shortages, ADA guidelines, United Kingdom healthcare policy

## Abstract

**Background:** Several GLP-1 receptor agonists (GLP-1 RAs) are used to treat type 2 diabetes (T2DM). Their cardio- and renal-protective effects and their association with substantial weight loss have been evident and progressively expanded their role in the American Diabetes Association (ADA) guidelines, which are endorsed by the European Association for the Study of Diabetes (EASD). The increased demand led to a global shortage. **Methods:** We utilized a repeated cross-sectional design, drawing data from national prescribing databases, to analyze six GLP-1 RAs: Dulaglutide, Exenatide, Liraglutide, Lixisenatide, Semaglutide, and Tirzepatide. AutoRegressive Integrated Moving Average (ARIMA) models with exogenous variables were applied to assess the trends over time and in different regions. **Results:** The prescription rates significantly differ between regions. Wales shows the highest prescribing rate for most GLP-1 RAs. The ARIMA models indicated a significant increase in their prescribing rates after the release of the 2022 ADA guidelines (e.g., Dulaglutide: Post-ADA effect of 15.22, 95% CI: [12.97, 17.47]). Following the GLP-1 RA shortages in July 2023, the prescribing rates, particularly for Semaglutide, increased (Shortage effect: 74.36, 95% CI: [71.92, 76.80]). Model diagnostics, including the Akaike Information Criterion (AIC) and Durbin–Watson statistics, confirmed the robustness of these trends. **Conclusions:** Informed decisions should be made by considering the prescribing trends before and after important events such as the issuing of new guidelines or safety alerts.

## 1. Introduction

Diabetes Mellitus (DM) affects around 530 million adults worldwide. The global prevalence is 10.5% among adults, with type 2 diabetes mellitus (T2DM) corresponding to around 98% of them [1]. Within the UK, the estimated prevalence of T2DM is 4.7 million, accounting for 7.4% of the population [2]. The rise in type 2 diabetes prevalence among young people in the UK affects their choice of long-term management, with around GBP 10.7 billion spent on treating diabetes and its complications annually [3]. The majority of the cost is directed to the treatment of complications and the associated morbidities. One of the closely associated comorbidities of T2DM is obesity. Around 65% of adults in the UK are overweight or obese, with the prevalence increasing [3,4,5,6]. The social annual cost to the UK is estimated to be GBP 58 billion, corresponding to 3% of its gross domestic product (GDP) [7].

Glucagon-like peptide-1 receptor agonists (GLP-1 RA), such as Exenatide, Liraglutide, Dulaglutide, and Semaglutide, are considered an evolution in the treatment of T2DM. They are involved in glucose metabolism by mimicking the action of the natural GLP-1 hormone. They also promote insulin secretion and inhibit glucagon secretion, hindering gastric motility [8]. GLP-1 RAs were initially introduced as a second- or third-line treatment for diabetic patients who failed to achieve their therapeutic goal with metformin, sulfonylureas, or dipeptidyl peptidase-4 (DPP-4) inhibitors [9]. During the previous decades, a paradigm shift in diabetes management occurred with the focus now on treating or avoiding complications [10]. GLP-1 RAs showed cardiovascular and renal benefits and weight reduction effects beyond their glycemic control effect, which led to them gaining prominence in the diabetes guidelines [11,12]. For example, in the 2022 American Diabetes Association (ADA) guidelines of standards of care in diabetes, the GLP-1 RAs benefits make them a preferred choice for treating T2DM in patients with additional complications or at risk for complications such as atherosclerotic cardiovascular disease, chronic kidney disease, and obesity [13]. The European Association for the Study of Diabetes (EASD) has endorsed these guidelines, and together, they published a consensus report [14]. In addition, studies revealed that T2DM patients prefer GLP-1 RAs over other antidiabetic medications, especially the less frequent dosing formulations [15]. Therefore, an increased demand for GLP-1 RAs is expected.

Several challenges face this demand, including high costs, limited insurance coverage, and shortages. Starting in 2022, the GLP-1 RA faced a global shortage due to a prescribing surge, manufacturing constraints, and spiked off-label use. The European Medicines Agency (EMA) and the Heads of Medicines Agencies (HMAs) have set recommendations via the Executive Steering Group on Shortages and Safety of Medicinal Products (MSSG) to monitor and implement plans for mitigating the shortages. In the United Kingdom (UK), a safety alert has been issued by the National Health Service (NHS), emphasizing prescribing GP-1 RAs for their licensed indications to overcome the global shortage.

Understanding the prescribing trends for GLP-1 RAs and the factors that affect them is crucial to providing insights about the integration of guidelines into clinical practice. It also assists future policy decisions to optimize the treatment of DM and obesity. Studies on the prescribing trends of GLP-1 RAs have been conducted in some countries [16,17,18]. None of these studies investigated the effect of the most recent guidelines or stock shortages on the prescribing trend. Similarly, no study describes the effect of these factors on the prescribing trends of GLP-1 RAs in the UK. Therefore, this study aimed to investigate the change in prescribing trends after both the ADA 2022 guidelines and the NHS shortage alert.

## 2. Materials and Methods

### 2.1. Study Design

This study investigated the GLP-1 RA prescribing trends in the UK from January 2018 to May 2024 using a repeated cross-sectional methodology. Due to data constraints, the analysis for England commenced in July 2019, while for Wales it began in April 2018. This study focused on the following GLP-1 RAs: Dulaglutide (BNF code: 0601023AQ), Exenatide (BNF code: 0601023Y0), Liraglutide (BNF code: 0601023AB), Lixisenatide (BNF code: 0601023AI), Semaglutide (BNF code: 0601023AW), and Tirzepatide (BNF code: 0601023AZ).

Data collection was conducted using publicly accessible resources. Prescribing data for England were retrieved from OpenPrescribing.net [19], data for Wales were from the NHS Wales Shared Services Partnership’s Prescribing Data Extracts [20], data for Scotland were from Public Health Scotland’s Monthly Prescribing Activity data [21], and data for Northern Ireland were from the GP Prescribing Data available on Open Data Northern Ireland [22]. All the data utilized in this study are openly available under the Open Government Licence (OGL) and did not require ethical approval. This study only examined prescriptions written by general practitioners (GPs) in community settings; prescriptions from hospitals or other healthcare facilities were not included. The extracted data include drug names, quantities, and prescribing dates.

### 2.2. Prevalence Calculation

To determine the prevalence of GLP-1 agonist prescriptions, the number of prescriptions per month was divided by the respective country’s population for that month and then multiplied by 100,000, yielding a standardized prescription rate per 100,000 individuals. Population data were obtained from official government websites. However, due to the unavailability of population data for Scotland and Northern Ireland for the years 2023 and 2024, as well as for all four countries in 2024, population figures for these periods were projected by applying the growth rate observed in the previous year.

### 2.3. Statistical Analysis

An analysis was conducted using AutoRegressive Integrated Moving Average (ARIMA) models with exogenous variables to assess the prescribing trends of GLP-1 receptor agonists throughout the United Kingdom. The investigation used monthly prescription data spanning several years and included Dulaglutide, Exenatide, Liraglutide, Lixisenatide, and Semaglutide. The statistical analysis tables did not include Tirzepatide because of its recent approval and the scarcity of prescription data. However, it was retained in the figure for the sake of completeness.

The dependent variable was the monthly prescriptions written for each medicine per 100,000 persons. The exogenous variables comprised the Northern Ireland, Scotland, and Wales country-specific indicators, with England serving as the reference group. The analysis also accounted for a baseline trend representing the prescribing behavior before the 2022 ADA guidelines, incorporating a six-month lag to reflect the changes in practice up to June 2022. A binary variable marked the period from July 2022 onwards, capturing the influence of the ADA guidelines. Additionally, the analysis included a trend for the period following the release of the ADA guidelines and a binary variable from July 2023 onwards, when a shortage of GLP-1 receptor agonists was observed. A post-shortage trend was also modeled.

Each medication was analyzed using an ARIMA (p, d, q) model, where ‘p’ denotes the autoregressive term, ‘d’ indicates the differencing needed to achieve stationarity, and ‘q’ represents the moving average term. The models were fitted using the Seasonal ARIMA with the exogenous factors (SARIMAX) function from the ‘statsmodels’ library, which allowed for the inclusion of the specified exogenous variables. The resulting coefficients and their associated 95% confidence intervals were reported. Model performance was assessed using the Akaike Information Criterion (AIC) and the Bayesian Information Criterion (BIC) to balance the model fit and complexity. The Durbin–Watson statistic was used to check for an autocorrelation in the residuals. Data cleaning and management were performed using Excel version 2409, while R version 4.4.0 was employed to run the statsmodels function and conduct the statistical analysis.

## 3. Results

Prescription patterns for the GLP-1 receptor agonists vary significantly across UK regions, as detailed in Table 1. Wales reports the highest average prescription rates for most GLP-1 RAs, including Semaglutide (179.52 per 100,000 population) and Dulaglutide (168.98 per 100,000 population). England shows moderate prescription rates, while Scotland and Northern Ireland generally have lower rates. Scotland leads in Exenatide prescriptions (17.4 per 100,000 population) despite a lower overall usage of the other GLP-1 RAs.

Figure 1 illustrates the temporal trends in GLP-1 RAs prescriptions, highlighting the effects of the ADA guideline update in January 2022 and the medication shortage in July 2023. Wales experienced the most significant increase in prescriptions following the ADA guidelines, especially for Semaglutide and Dulaglutide, with Semaglutide usage further surging during the shortage. In contrast, Scotland and Northern Ireland showed more muted responses, with Scotland even recording a decline in certain prescriptions, such as Liraglutide, post-shortage.

The ARIMA models (Table 2, Figure 2 and Figure 3) provide deeper insights into these trends, capturing the regional effects and temporal events’ impact. Significant coefficients are reported with 95% confidence intervals (CI95%).

The analysis of GLP-1 RAs prescribing across different regions in the UK revealed notable variations. In Scotland, Dulaglutide usage was significantly lower compared to England, with a coefficient of −39.94 (CI95%: [−42.56, −37.32]). Conversely, Wales exhibited a marked increase in prescriptions relative to England, with a coefficient of 40.81 (CI95%: [38.29, 43.33]), indicating a substantial difference. Northern Ireland, on the other hand, showed a modest decrease with a coefficient of −11.08 (CI95%: [−13.52, −8.64]). For Liraglutide, the effect was more significant in Wales, with a coefficient of 18.87 (CI95%: [16.59, 21.15]). Semaglutide prescribing was particularly higher in Wales, reflected in a coefficient of 69.80 (CI95%: [67.12, 72.48]), while Scotland reported lower prescribing rates with a coefficient of −32.42 (CI95%: [−34.50, −30.34]). Northern Ireland displayed a slight positive coefficient of nine (CI95%: [6.85, 11.15]) for Semaglutide.

The baseline trend analysis offered insights into the pre-existing trends before the influence of significant events like the ADA guideline update and medication shortages. Dulaglutide showed a positive baseline trend, with a coefficient of 14.36 (CI95%: [12.12, 21.60]), suggesting a steady increase in its prescribing over time. Semaglutide exhibited an even stronger baseline trend, indicated by a coefficient of 24.61 (CI95%: [22.05, 27.17]). In contrast, Liraglutide and Lixisenatide had either flat or negative baseline trends with coefficients of 2.02 (CI95%: [−0.53, 4.57]) and −2.31 (CI95%: [−4.57, 2.24]), respectively (Table 2).

The post-ADA guideline effect, measured with a 6-month lag after January 2022, was most pronounced for Semaglutide, which saw a significant increase in prescribing, as indicated by a coefficient of 18.24 (CI95%: [15.95, 20.53]). Dulaglutide also experienced a notable increase, with a coefficient of 15.22 (CI95%: [12.97, 17.47]), while other drugs such as Exenatide showed minimal or negative changes (Table 2).

Further analysis of the post-ADA trend highlighted the ongoing influences of the guidelines on drug prescriptions. Both Dulaglutide and Semaglutide exhibited positive trends; however, the confidence intervals included zero, indicating that these trends were not statistically significant. Liraglutide and Lixisenatide, in contrast, displayed minimal changes (Table 2).

The medication shortage effect was another critical factor, particularly impacting Semaglutide, which saw a sharp increase of 74.36 (CI95%: [71.92, 76.80]). Dulaglutide also experienced a significant positive effect, with a coefficient of 29.14 (CI95%: [26.72, 31.56]), while Liraglutide encountered a substantial negative impact, with a coefficient of −28.02 (CI95%: [−30.41, −25.63]). The post-shortage trend analysis explored the long-term effects of the medication shortage on usage patterns. Semaglutide continued to exhibit a positive trend, with a coefficient of 27.3 (CI95%: [15.1, 39.6]). Dulaglutide and Liraglutide showed minimal or negative trends (Table 2).

Lastly, the model quality was assessed using the Durbin–Watson statistic, AIC, and BIC. The Durbin–Watson statistics for all the models were close to two, indicating a minimal autocorrelation in the residuals and suggesting that the models are well specified. The AIC and BIC values varied, with Dulaglutide showing an AIC of 2526.57 and a BIC of 2566.71, indicating a relatively good model fit. Semaglutide, with an AIC of 2410.62 and a BIC of 2448.77, demonstrated a robust model with strong predictive power. These indicators confirm the adequacy of the models in capturing the underlying data structures, making them reliable tools for understanding the trends and their influence on GLP-1 receptor agonist usage across different regions and time periods.

## 4. Discussion

This study offers a comprehensive analysis of the temporal and regional variation shifts in the prescribing patterns of GLP-1 receptor agonists across the United Kingdom. It mainly highlights the effect of the ADA guidelines and drug shortages on these patterns. Semaglutide and dulaglutide were the most prescribed drugs during the study period. The results show a significant increase in prescriptions following the endorsement of the ADA guidelines, particularly for Semaglutide, Dulaglutide, and Liraglutide. Their prescribing trends have insignificantly increased. A study in the United States showed a fast increase in the prescribing rate of GLP-1 RAs in the period of 2012–2022 [16]. Another study in Australia demonstrated a dramatic rise in GLP-1 RAs use from 2014 to 2022 [23]. The observed rapid increase in prescriptions represents an apparent shift in the prescribing pattern of this therapeutic category, which will probably continue to accelerate as data on the potential cardiovascular advantages and positive impacts on weight loss outcomes become available. The analysis of the GIP/GLP-1 receptor agonist Tirzapatide prescribing rate was restricted to a 12-month period. However, it seems that it will continue to accelerate with a surge comparable to the one seen with the other drugs. Regarding health outcomes, GLP-1 RAs are associated with the control of blood glucose levels and reduced cardiovascular events and strokes, which could translate into cost-saving benefits [24]. A comparative meta-analysis showed that the GLP-1 RAs improved blood pressure, lipid, and glycemia profiles compared to insulins in T2DM patients [25]. GLP-1 RAs may also help patients avoid surgical weight loss interventions and reduce their medical costs. During the study period, Lixisenatide was the least prescribed. This may be attributed to the more significant clinical benefits and reduced cost of Semaglutide compared to Lixisenatide [26,27]. In addition, the once-weekly regimen may contribute to patients’ preferences for these drugs [15]. This attribute may also be the driving force behind slowing the prescribing rate of Liraglutide over time.

In Wales, the GLP-1 RA prescribing rate surpassed those of other regions, while in Scotland, there was a marked decrease in the prescribing rate. This could be explained by the marked increase in the obesity rate among Wales adults compared to the other regions [28]. Another possible explanation is the higher prevalence of diabetes among the Walsh population compared to the population in Scotland [29,30]. This approach in Scotland is aligned with the conservative recommendations for adopting GLP-1 RAs taken from the NG28 NICE guidelines for diabetes [31], where GLP-1 RAs remain a fourth line of treatment, which is much later in the treatment algorithm than it is in the American guidelines from the ADA [10]. Additionally, the NG28 recommends a discontinuation of GLP-1 RAs after six months if there is no significant reduction in the A1c and the weight [31].

In July 2023, the NHS issued a safety alert regarding GLP-1 RA shortages, requiring actions for clinicians and prescribers to emphasize prescribing these drugs in their licensed indications and forbidding the initiation of new patients on these drugs. The NHS also recommends against prescribing excessive GLP-1 RAs [32]. After this safety alert, Liraglutide prescriptions were significantly reduced. Semaglutide and Dulaglutide prescriptions significantly increased, with an insignificant increase in prescribing rates. One possible explanation is that the NHS later issued an update that superseded the previous one but emphasized that the supply would continue to be limited until the end of 2024. However, the safety alert mentioned that Semaglutide tablets would be available in sufficient quantities. It is possible that the prescribers switched from Liraglutide to alternatives like Semaglutide.

Drug shortages are a significant problem in providing healthcare to the community [33]. They have become a global phenomenon that is continuously rising [29,30,31]. In the UK, the Department of Health and Social Care (DHSC) received 137 notifications of drug shortages on average each month in 2023 [34]. The British Generic Manufacturers Association (BGMA) revealed that drug shortages have increased by 100% from January 2022 to January 2024 [35]. This emphasizes the need for informed decisions in accordance with the prescribing behaviors before and after issuing safety alerts.

In conclusion, this study has shown an increase in the prescribing of GLP-1 RAs in the UK and the regional differences in these trends in response to the ADA guidelines in 2022, emphasizing the impact of international guidelines on the prescribing patterns. It also demonstrated the change in prescribing trends after the medication shortage alert in different regions. These findings highlight the need for informed decisions through close monitoring of the prescribing behaviors before and after important international events. For example, healthcare policymakers may expect and avoid drug shortages and ensure the consistency and equitability of medication access across all geographical regions. Further studies are needed to explore the underlying causes of these regional differences in response to the international guidelines and medication shortage alerts.

Limitations: This study encountered several limitations. First, the data represent prescribing in community settings rather than hospital facilities, which may affect the comprehensiveness of the findings. The datasets used in this study were sourced from open databases that do not contain detailed patient demographics or specific comorbidities. The ARIMA models do not completely account for unpredicted drops and spikes because they rely on the stationarity assumption. However, the use of ARIMA remains appropriate for the majority of our analysis, as the primary focus was on identifying underlying trends and patterns, rather than isolated anomalies. The recent introduction of Tirzepatide to the UK market limited the ability to evaluate its trends fully.

As for the ethical considerations, data collection was conducted using publicly accessible resources and data were openly available under the Open Government Licence (OGL) and did not require ethical approval for use.

## Figures and Tables

**Figure 1 jcm-13-06256-f001:**
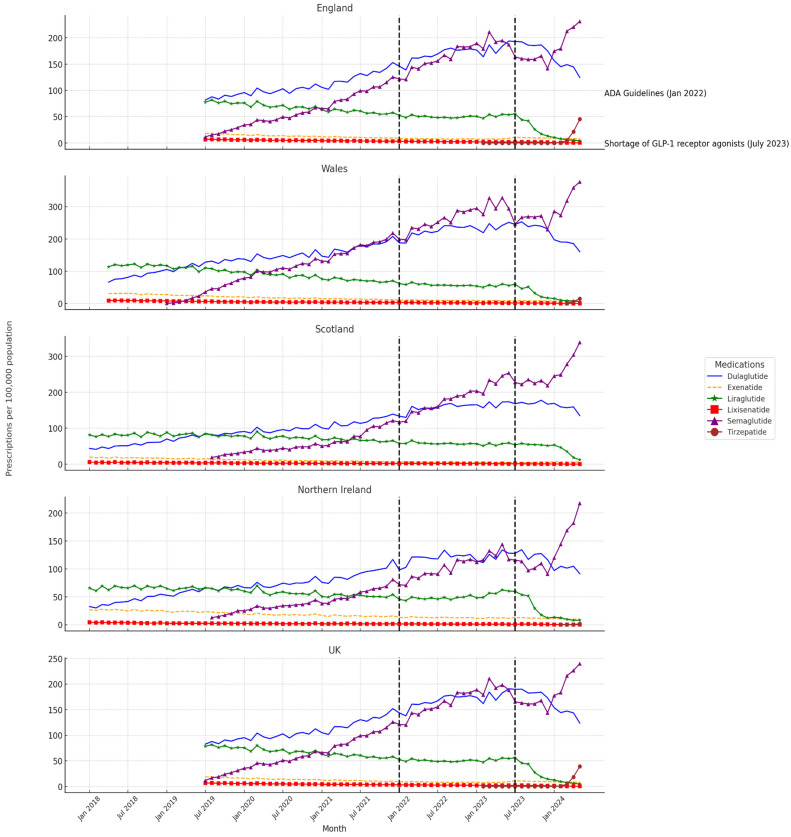
The trends in GLP-1 Receptor Agonist prescriptions across the UK regions including the impact of the ADA Guidelines (January 2022) and the July 2023 medication shortage. This set of line graphs illustrates the prescription trends of various Glucagon-like peptide-1 receptor agonists (GLP-1 RA) across the UK. The data span from January 2018 to May 2024 for Wales, Scotland, and Northern Ireland, and from July 2019 to May 2024 for England and the UK as a whole. Each medication is represented by a distinct colored line: Dulaglutide (blue), Exenatide (dashed yellow), Liraglutide (green), Lixisenatide (red), Semaglutide (purple), and Tirzepatide (brown). The graphs include two vertical dashed black lines marking the following significant events: the issuance of the ADA guidelines in January 2022 and the onset of a medication shortage in July 2023, illustrating their impact on prescribing patterns.

**Figure 2 jcm-13-06256-f002:**
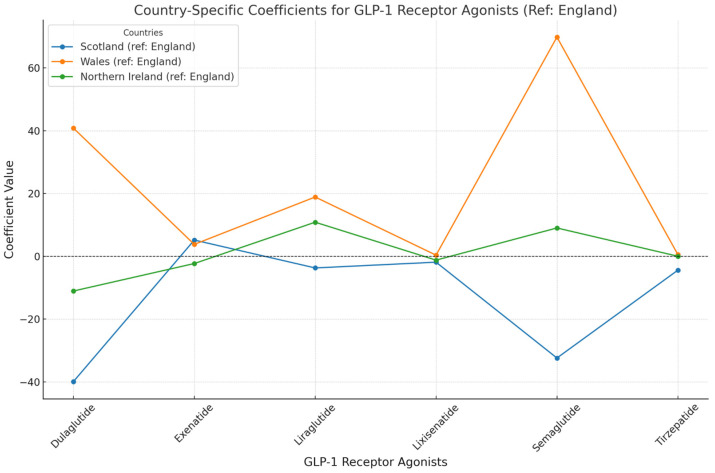
The trends in GLP-1 Receptor Agonist prescriptions across the UK Regions: the impact of the ADA Guidelines (January 2022). This line graph depicts the country-specific coefficients for prescriptions of different GLP-1 RAs across Scotland, Wales, and Northern Ireland using England as a reference. The *x*-axis represents the different GLP-1 RAs (Dulaglutide, Exenatide, Liraglutide, Lixisenatide, Semaglutide, and Tirzepatide). The *y*-axis indicates the coefficient values, which reflect the degree to which the prescriptions differ from those in England. Each country is represented by a distinct colored line: Scotland (blue), Wales (orange), and Northern Ireland (green). Points above the horizontal zero line indicate a higher prescribing rate compared to England, while points below suggest a lower rate.

**Figure 3 jcm-13-06256-f003:**
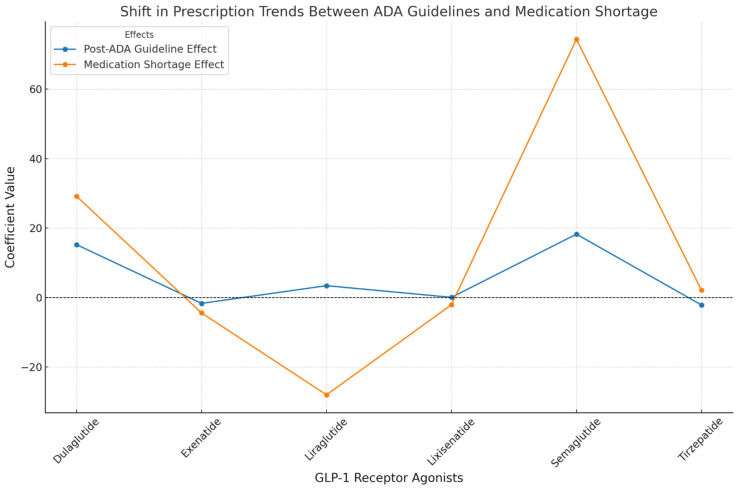
The comparative impact of ADA Guideline updates and the medication Shortage on the prescription trends for GLP-1 Receptor Agonists. This line graph illustrates the coefficient values for the shift in prescription trends for GLP-1 RAs following the implementation of the ADA guidelines in Jan 2022 (blue line) and the subsequent medication shortage in July 2023 (orange line).

**Table 1 jcm-13-06256-t001:** Mean and standard deviation of GLP-1 Receptor Agonist prescriptions per 100,000 population by country.

Country	Number of Prescriptions per 100,000 Population					
	Dulaglutide Mean (SD)	Exenatide Mean (SD)	Liraglutide Mean (SD)	Lixisenatide Mean (SD)	Semaglutide Mean (SD)	Tirzepatide Mean (SD)
England	138.04 (35.92)	10.75 (3.12)	53.43 (19.77)	3.37 (1.8)	115.28 (63.44)	4.49 (12.1)
Northern Ireland	114.1 (42.42)	9.92 (4.83)	67.23 (14.71)	2.55 (1.33)	132.85 (87.65)	None
Scotland	85.86 (30.08)	17.4 (5.36)	52.75 (15.3)	1.85 (0.97)	77.55 (45.79)	0.52 (0.71)
Wales	168.98 (53.12)	15.94 (7.5)	75.35 (30.86)	4.16 (2.5)	179.52 (101.17)	7.08 (7.29)

**Table 2 jcm-13-06256-t002:** ARIMA Model coefficients for GLP-1 Receptor Agonists: regional effects and impacts of ADA Guidelines and medication shortages.

Coefficient *	Dulaglutide	Exenatide	Liraglutide	Lixisenatide	Semaglutide	All GLP-1s **
Durbin–Watson	1.87	2.05	2.03	2.02	1.92	1.99
AIC	2526.57	1469.54	2246.11	861.85	2410.62	1848.98
BIC	2566.71	1509.67	2286.24	901.99	2448.77	1982.92
Scotland Coefficient (ref: England)	**−39.94 [−42.56, −37.32]**	**5.15 [2.78, 7.52]**	**−3.73 [−5.10, −2.36]**	**−1.90 [−3.42, −0.38]**	**−32.42 [−34.50, −30.34]**	**−12.72 [−14.89, −10.55]**
Wales Coefficient (ref: England)	**40.81 [38.29, 43.33]**	**3.77 [1.55, 6.00]**	**18.87 [16.59, 21.15]**	0.34 [−2.12, 2.80]	**69.80 [67.12, 72.48]**	**22.03 [19.78, 24.28]**
Northern Ireland Coefficient (ref: England)	**−11.08 [−13.52, −8.64]**	−2.33 [−4.67, 0.01]	**10.84 [8.62, 13.06]**	−1.24 [−3.59, 1.11]	**9.00 [6.85, 11.15]**	0.87 [−1.04, 2.78]
Baseline Trend	**14.36 [12.12, 21.60]**	**6.10 [3.82, 8.38]**	2.02 [−0.53, 4.57]	−2.31 [−4.57, 2.24]	**24.61 [22.05, 27.17]**	**17.37 [12.82, 21.92]**
Post-ADA Guideline Effect (lag time of 6 months after Jan 2022)	**15.22 [12.97, 17.47]**	−1.68 [−4.12, 0.76]	**3.41 [0.95, 5.87]**	0.05 [−2.39, 2.49]	**18.24 [15.95, 20.53]**	**5.02 [2.85, 7.19]**
Post-ADA Trend (lag time of 6 months after Jan 2022)	5.1 [−4.0,14.2]	0.31 [−3.0,3.6]	2.2 [−1.5,5.9]	0.04 [−2.0,2.1]	6.2 [−3.0,15.4]	3.4 [−2.5,9.3]
Medication Shortage Effect	**29.14 [26.72, 31.56]**	**−4.42 [−6.85, −1.99]**	**−28.02 [−30.41, −25.63]**	−2.07 [−4.48, 0.34]	**74.36 [71.92, 76.80]**	**11.19 [8.42, 13.96]**
Post-Shortage Trend	7.1 [−5.0,19.2]	−1.5 [−5.0,2.0]	−12.4 [−18.0,−7.0]	−0.21 [−4.0,3.6]	27.3 [15.1,39.6]	4.6 [−2.0,11.2]

* Tirzepatide could not be calculated due to its recent approval, resulting in insufficient data. ** This includes all GLP-1s with the exception of Tirzepatide. The significant value is highlighted in bold.

## Data Availability

This study’s data analysis is based solely on publicly available sources. The prescribing data for England were sourced from OpenPrescribing.net, data for Wales were from the NHS Wales Shared Services Partnership’s Prescribing Data Extracts, data for Scotland were from Public Health Scotland’s Monthly Prescribing Activity data, and data for Northern Ireland were from the GP Prescribing Data available on Open Data Northern Ireland. All data utilized are available under the Open Government Licence (OGL) and can be accessed directly through these platforms.

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
