# Peer review of "Impact of ADA Guidelines and Medication Shortage on GLP-1 Receptor Agonists Prescribing Trends in the UK: A Time-Series Analysis with Country-Specific Insights"

_jcm, 2024, doi:10.3390/jcm13206256_

Round 1
Reviewer 1 Report
Comments and Suggestions for Authors
The main question addressed, originality and relevance in the field
The primary issue in this study was whether UK prescribing trends were impacted by both the 2022 American Diabetes Association (ADA) recommendations and the absence of GLP-1 receptor agonism (GLP-1 RA) in July 2023. This work addresses an important and topical issue: the effect of new clinical recommendations and medication shortages on prescription habits. Furthermore, focusing on medications such as Dulaglutide, Semaglutide, and Liraglutide, this study investigates geographical variations in GLP-1 RA prescription trends.
Specific improvements for methodology:
The method uses a repeated cross-sectional design and ARIMA models for trend analysis, which are suitable for the available data, but the authors could improve robustness by adding patient demographics such as age, sex, and comorbidities that could explain prescribing trends. It would be easier to understand why prescribing rates differ in certain regions if patient demographics were included.
Consistency of conclusions:
A conclusions section that includes future perspectives and clear evidence of regional differences in prescribing trends is needed.
References:
References are adequate and cover relevant directions, previous studies on GLP-1 RA and global trends in diabetes.
Further comments on the tables and figures:
The authors could consider improving the readability of the figures by including more detailed legends, and directly explaining any trends or important points in the figure descriptions, rather than just at the beginning of the Results section.
Author Response
The main question addressed, originality and relevance in the field
The primary issue in this study was whether UK prescribing trends were impacted by both the 2022 American Diabetes Association (ADA) recommendations and the absence of GLP-1 receptor agonism (GLP-1 RA) in July 2023. This work addresses an important and topical issue: the effect of new clinical recommendations and medication shortages on prescription habits. Furthermore, focusing on medications such as Dulaglutide, Semaglutide, and Liraglutide, this study investigates geographical variations in GLP-1 RA prescription trends.
Specific improvements for methodology:
Comment 1: The method uses a repeated cross-sectional design and ARIMA models for trend analysis, which are suitable for the available data, but the authors could improve robustness by adding patient demographics such as age, sex, and comorbidities that could explain prescribing trends. It would be easier to understand why prescribing rates differ in certain regions if patient demographics were included.
Response 1: Thank you for your thoughtful feedback on our methodological approach. We appreciate your suggestions. However, these datasets have been extracted from open databases, and unfortunately, they don’t contain patients' demographics or details. We have mentioned this limitation in the discussion to ensure transparency about the constraints of our analysis (Line # 318). Stated in the manuscript as follows: “The datasets used in this study were sourced from open databases that do not contain detailed patient demographics or specific comorbidities.”
Comment 2: Consistency of conclusions:
A conclusions section that includes future perspectives and clear evidence of regional differences in prescribing trends is needed.
Response 2: Thank you for your comment. We have added a conclusion section (Line# 306)
Stated in the manuscript as follows
“In conclusion, this study has shown an increase in the prescribing of GLP-1 RA in the UK and the regional differences in these trends in response to the ADA guidelines in 2022, emphasizing the impact of international guidelines on the prescribing patterns. It also demonstrated the change in prescribing trends after the medication shortage alert in different regions. These findings highlighted the need for informed decisions by closely monitoring the prescribing behaviors before and after important international events. For example, healthcare policymakers may expect and avoid drug shortages and ensure consistency and equitability of medication access across all geographical regions. Further studies are needed to explore the underlying causes of these regional differences in response to international guidelines and medication shortage alerts.”
Comment 3: References:
References are adequate and cover relevant directions, previous studies on GLP-1 RA and global trends in diabetes.
Response 3: Thank you for your positive evaluation.
Comment 4: Further comments on the tables and figures:
The authors could consider improving the readability of the figures by including more detailed legends, and directly explaining any trends or important points in the figure descriptions, rather than just at the beginning of the Results section.
Response 4: Thank you for your valuable comment. We have added an explanation in the legends of the figures as follows.
Figure 1. Trends in GLP-1 Receptor Agonist Prescriptions Across the UK Regions: Impact of ADA Guidelines (January 2022) and July 2023 Medication Shortage.
This set of line graphs illustrates the prescription trends of various Glucagon-like peptide-1 receptor agonists (GLP-1 RA) across the UK. The data spans from January 2018 to May 2024 for Wales, Scotland, and Northern Ireland, and from July 2019 to May 2024 for England and the UK as a whole. Each medication is represented by a distinct colored line: Dulaglutide (blue), Exenatide (dashed yellow), Liraglutide (green), Lixisenatide (red), Semaglutide (purple), and Tirzepatide (brown). The graphs include two vertical dashed black lines marking significant events: the issuance of the ADA guidelines in January 2022 and the onset of a medication shortage in July 2023, illustrating their impact on prescribing patterns.
Figure 2. Trends in GLP-1 Receptor Agonist Prescriptions Across the UK Regions: Impact of ADA Guidelines (January 2022).
This line graph depicts the country-specific coefficients for prescriptions of different GLP-1 RAs across Scotland, Wales, and Northern Ireland using England as a reference. The x-axis represents different GLP-1 RAs (Dulaglutide, Exenatide, Liraglutide, Lixisenatide, Semaglutide, and Tirzepatide). The y-axis indicates the coefficient values, which reflect the degree to which prescriptions differ from those in England. Each country is represented by a distinct colored line: Scotland (blue), Wales (orange), and Northern Ireland (green). Points above the horizontal zero line indicate a higher prescribing rate compared to England, while points below suggest a lower rate.
Figure 3. Comparative Impact of ADA Guideline Updates and Medication Shortage on Prescription Trends of GLP-1 Receptor Agonists.
This line graph illustrates the coefficient values for the shift in prescription trends of GLP-1 RAs following the implementation of ADA guidelines in Jan 2022 (blue line) and the subsequent medication shortage in July 2023 (orange line).
Reviewer 2 Report
Comments and Suggestions for Authors
The authors conducted a repeated cross-sectional analysis using data from national prescribing databases to analyze six GLP-1 RAs to assess the trends of prescription rates over time and regions in the UK. In the study, it was found that the prescription rates were significantly different between regions. The model showed a significant increase in prescription rates after ADA guidelines 2022. After the showtages of GLP-1 RA in July 2023, there was some reactions in the prescribing rates in some GLP-1 RAs. Based on the results, the authors concluded that informed decision should be made by considering prescribing trends before and after important events such as the issuing new guidelines or safety alerts.
This is an intersting article showing how the prescription rates were changed following important events such as issuing of the guidelines and the medication shortage. I have some comments for consideration:
- In line 16 in the abstract, the word "Enatide" should be "Exenatide".
- In line 87-92 in the Materials and Methods, the authors described the data source. This section should be further characterized for better understanding of the reader. For instance, which variables are available (e.g., demographics, date of despensing, strength, drug name, duration of prescription, and calender years covered by each data source).
- Further to this section, in this study, several different data extracts were used to cover different regions in the UK. Are there any heterogeneity that should be explained. For instance, are there differences in data collection settings (primary care or hospital settings) and health insurance/reimbursement system among data sources?
- In line 130 in Statistical Analysis, the version of R software should be included in the main text.
- The authors concluded that informed decision should be made by considering prescribing trends before and after important events such as the issuing new guidelines or safety alerts. The connection between the study findings and this conclusion is not clear. It is recommended that authors discuss further in the article that how the reactions of prescription rates necessitates the informed decision making considering the trends in prescription rates before and after important events.
Comments on the Quality of English Language
No further comments.
Author Response
Comment 1: The authors conducted a repeated cross-sectional analysis using data from national prescribing databases to analyze six GLP-1 RAs to assess the trends of prescription rates over time and regions in the UK. In the study, it was found that the prescription rates were significantly different between regions. The model showed a significant increase in prescription rates after ADA guidelines 2022. After the showtages of GLP-1 RA in July 2023, there was some reactions in the prescribing rates in some GLP-1 RAs. Based on the results, the authors concluded that informed decision should be made by considering prescribing trends before and after important events such as the issuing new guidelines or safety alerts.
This is an intersting article showing how the prescription rates were changed following important events such as issuing of the guidelines and the medication shortage. I have some comments for consideration:
- In line 16 in the abstract, the word "Enatide" should be "Exenatide".
Response 1: Thank you for your thoughtful feedback on our manuscript. Concerning the highlighting typo, we have corrected it in line#16.
- Comment 2: In line 87-92 in the Materials and Methods, the authors described the data source. This section should be further characterized for better understanding of the reader. For instance, which variables are available (e.g., demographics, date of despensing, strength, drug name, duration of prescription, and calender years covered by each data source).
Response 2: We agree that more explanation should be added. The Materials and Methods section has been updated with lines 87–92 to contain information on the available variables. The extracted data include drug names, quantities, and prescribing dates. There are no other demographic data available in these databases.
- Comment 3: Further to this section, in this study, several different data extracts were used to cover different regions in the UK. Are there any heterogeneity that should be explained. For instance, are there differences in data collection settings (primary care or hospital settings) and health insurance/reimbursement system among data sources?
Response 3: Thank you for your comment. All the extracted data represent primary care settings, as we mentioned in the methodology section (Line 93), stated in the manuscript as follows: “This study only examined prescriptions written by general practitioners (GPs) in community settings”. Prescriptions from hospitals or other healthcare facilities were not included.
Concerning the insurance/reimbursement systems among the four countries. Wales, Scotland, and Northern Ireland have made prescriptions free for all residents. In England, a fixed charge per prescription item is charged.
For evidence and details, here is references:
https://www.nhs.uk/nhs-services/prescriptions/nhs-prescription-charges/
https://www.nhsinform.scot/care-support-and-rights/nhs-services/pharmacy/prescription-charges-and-exemptions/
We compared the prescription trends among the regions using England as a reference. Wales recorded high prescription rates, while Scotland recorded lower than England's. So, we believe that reimbursement policies may not solely influence the differences in prescription rates across the UK regions. However, we further emphasize the need for comprehensive studies considering a broader range of factors on prescribing patterns and their regional variations in our conclusion.
- Comment 4: In line 130 in Statistical Analysis, the version of R software should be included in the main text.
Response 4: As suggested, we have included the specific version of the R (4.4.0) software used for our statistical analysis. This is updated in line 130 of the Statistical Analysis section.
This is stated in the manuscript as follows, “The Durbin-Watson statistic was used to check for autocorrelation in the residuals. Data cleaning and management were performed using Excel, while (R version 4.4.0) was employed to run the statsmodels function and conduct the statistical analysis”
- Comment 5: The authors concluded that informed decision should be made by considering prescribing trends before and after important events such as the issuing new guidelines or safety alerts. The connection between the study findings and this conclusion is not clear. It is recommended that authors discuss further in the article that how the reactions of prescription rates necessitates the informed decision making considering the trends in prescription rates before and after important events.
Response 5: Thank you for your valuable comment. Indeed, more clarification is needed regarding the informed decision-making statement. So, we have expanded the Discussion section and added a conclusion explaining this statement.
This is stated in the manuscript as follows:
“In conclusion, this study has shown an increase in the prescribing of GLP-1 RA in UK and the regional differences in these trends in response to the ADA guidelines in 2022, emphasizing the impact of international guidelines on the prescribing patterns. It also demonstrated the change in prescribing trends after the medication shortages in different regions. These emphasize the need for informed decisions by closely monitoring the prescribing behaviors before and after important international events. For example, healthcare policymakers may expect and avoid drug shortages, and ensure consistency and equitability of medication access across all regions in the UK. Further studies are needed to explore the underlying causes of these regional differences in response to international guidelines and medication shortage alerts.”
Round 2
Reviewer 1 Report
Comments and Suggestions for Authors
Accept in present form.
Reviewer 2 Report
Comments and Suggestions for Authors
The authors have adequately addressed my comments. I have no further comments.